# Metabolic Alterations in Canine Mammary Tumors

**DOI:** 10.3390/ani13172757

**Published:** 2023-08-30

**Authors:** Guilherme Henrique Tamarindo, Adriana Alonso Novais, Luiz Gustavo Almeida Chuffa, Debora Aparecida Pires Campos Zuccari

**Affiliations:** 1Department of Molecular Biology, São José do Rio Preto Faculty of Medicine, São José do Rio Preto 15090-000, SP, Brazil; 2Brazilian Biosciences National Laboratory, Brazilian Center for Research in Energy and Materials (CNPEM), Campinas 13083-970, SP, Brazil; 3Health Sciences Institute (ICS), Mato Grosso Federal University (UFMT), Sinop 78550-728, MT, Brazil; 4Department of Structural and Functional Biology, Institute of Biosciences, São Paulo State University (UNESP), Botucatu 18618-689, SP, Brazil

**Keywords:** canine mammary tumors, metabolism, cancer, mitochondria, metabolic reprogramming, glucose, amino acids, lipids

## Abstract

**Simple Summary:**

Cancer cells usually have a short timeframe for proliferation, which favors tumor growth. Therefore, they require more energy and intermediates to sustain biosynthetic pathways that will supply all the requirements for cell division. This event is known as metabolic reprogramming and is described in all cancer types, it also being a vulnerability for therapy. However, metabolic alterations in canine mammary tumors are poorly explored. In this review, we compile the metabolic rewiring described in canine mammary tumors, which could be used as a therapeutic opportunity for treatment in veterinary oncology.

**Abstract:**

Canine mammary tumors (CMTs) are among the most common diseases in female dogs and share similarities with human breast cancer, which makes these animals a model for comparative oncology studies. In these tumors, metabolic reprogramming is known as a hallmark of carcinogenesis whereby cells undergo adjustments to meet the high bioenergetic and biosynthetic demands of rapidly proliferating cells. However, such alterations are also vulnerabilities that may serve as a therapeutic strategy, which has mostly been tested in human clinical trials but is poorly explored in CMTs. In this dedicated review, we compiled the metabolic changes described for CMTs, emphasizing the metabolism of carbohydrates, amino acids, lipids, and mitochondrial functions. We observed key factors associated with the presence and aggressiveness of CMTs, such as an increase in glucose uptake followed by enhanced anaerobic glycolysis via the upregulation of glycolytic enzymes, changes in glutamine catabolism due to the overexpression of glutaminases, increased fatty acid oxidation, and distinct effects depending on lipid saturation, in addition to mitochondrial DNA, which is a hotspot for mutations. Therefore, more attention should be paid to this topic given that targeting metabolic fragilities could improve the outcome of CMTs.

## 1. Introduction

Canine mammary tumors (CMTs) are the most common cancer in female dogs that have not been surgically neutered [1], accounting for almost 50% of all canine neoplasms [2]. They arise spontaneously with increased risk through aging [3], mainly between 8 and 11 years, and also in certain breeds [4]. The deregulation of sexual hormones, such as from exposure to endogenous ovarian hormones, may cause the development of mammary tumors in dogs [5], but other factors may also influence CMTs, such as obesity in the early stages of life [6], inflammation [7], and an increase in free radicals and reactive oxygen species (ROS) [8]. CMTs may appear as single or multiple nodules, and posterior mammary glands are more frequently affected than anterior glands [9]. Approximately 50% of CMTs are malignant [10], with the most common tumor type among them being tubular carcinoma (adenocarcinoma), followed by papillary, solid, and complex carcinomas in addition to carcinosarcoma. The other half is benign, with fibroadenomas, ductal papillomas, benign mixed tumors, and simple adenomas being the most common [11]. Despite such classifications, it is also common to find more than one tumor type in different mammary glands in the same patient [12], and CMTs upon metastasis usually show tropism to the lymph node areas and lungs [6]. In the present review, we emphasized malignant tumors, unless otherwise stated, referred to here as CMTs. Also, CMTs are associated in the literature with stages, grades, and subtypes. “Stage” is a medical term adopted to assess the extent of cancer. In the specific context of mammary tumors in dogs, staging follows the TNM system, where T signifies the clinical tumor size using thresholds like 3 and 5 cm; N indicates nodal metastasis presence; and M denotes distant metastasis diagnosed through palpation, medical imaging, biopsy, or cytology [10,13]. Conversely, grade refers to a classification that assesses the appearance of cancer cells under a microscope compared with the normal cells of the same tissue. The cancer grade indicates how abnormal the cells appear and how quickly they can multiply and spread. Typically, grades are assigned on a numerical scale (e.g., from 1 to 3) or using descriptive terms (well-differentiated, moderately differentiated, poorly differentiated). A lower grade usually indicates cancer cells that resemble normal cells more and tend to grow more slowly, while a higher grade indicates more abnormal cells that are more prone to rapid growth and spread. In addition, “subtype” refers to the histological status via estrogen receptor (ER), progesterone receptor (PR), and human epidermal growth factor receptor 2 (HER2/ERRB2) expression.

CMTs are the leading cause of death in aged female dogs and have a higher incidence compared with human breast cancer, mainly in low-income countries [2,14]. CMTs are highly prevalent in non-spayed female dogs, but a significant proportion of animals start with premalignant lesions that can progress to invasive cancer within a relatively short period of time. This scenario of high incidence and, sometimes, poor prognosis, highlights the need for new assertive therapeutic approaches. A remarkable alteration commonly documented among all cancers is metabolic reprogramming, which supplies the bioenergetic demands and the anabolic requirements for cell proliferation and tumor growth. Although it has been widely investigated in human breast cancer, there is a lack of information on metabolic rewiring in CMTs. In this dedicated review, we summarize metabolic adjustments described in the literature, which could serve as a therapeutic strategy with a focus on lipid, glucose, and amino acid metabolism.

## 2. Overview of Metabolic Features of Mammary Cancer

Cancer cells undergo several alterations to reach an imbalance between a higher proliferative rate and cell death, providing a molecular context whereby neoplastic cells promote tumor growth and even metastasis. Among such adjustments is metabolic reprogramming, one of the hallmarks of carcinogenesis, which, for instance, meets the bioenergetic and biosynthetic demands of cell proliferation [15,16,17]. In 1956, Otto Warburg described how tumor cells uptake more glucose and use it as an energy source through aerobic glycolysis because of the impairment of mitochondria [18]. This was later revisited, and it was demonstrated that mitochondrial activity is decreased and not impaired, different from what was originally proposed as the Warburg effect [19]. The increase in glycolysis, a very inefficient ATP source compared with oxidative phosphorylation (OXPHOS) [20], provides energy production but also intermediates for anabolic pathways such as nucleic acid synthesis and NADPH, all of them required for cell division and tumor growth [19]. Also, glutamine is another carbon source that fuels the tricarboxylic acid cycle (TCA) via reductive carboxylation [21], which is often necessary for cell survival, mostly in advanced cancers [21,22,23]. Glutamine uptake and consumption increase with oxidative stress [24,25] and may be used for fatty acid synthesis, which, in turn, provides phospholipids to membrane synthesis, required for cell cycle progression [26]. In addition to glucose and glutamine, lipids are also crucial for cancer cell survival and progression to more aggressive phenotypes [27,28,29,30]. Lipids are the building blocks of membrane synthesis, post-translational modification, and energy generation. It was reported that endogenously synthesized fatty acids are required for cancer cell proliferation [26], and the pharmacological inhibition of fatty acid synthase (FASN) has antitumor effects in cell lines, organoids, and xenograft models [29]. Interestingly, not only endogenous but also exogenous fatty acids obtained through diet have been shown to modulate cancer progression [31], including through the regulation of metabolism [32]. Taken together, this short summary highlights how the rewiring of metabolism is a vulnerability that has been explored as a therapeutic opportunity in several cancers.

In the literature, most available information on tumor metabolism has been generated from human breast cancer, but given the similarities with CMT, it is reasonable to consider whether some of these alterations can be found in dogs. In breast cancer, tumor cells display high glucose uptake compared with normal ones [33], and the former may regulate carbohydrate availability to favor metastasis [34]. Breast cancer cells have glucose transporter (GLUT) overexpression [35], GLUT1 being related to poor prognosis and higher proliferation. However, GLUT overexpression has distinct patterns in malignant human cell lines [35,36], suggesting that, although glucose uptake increases, there are adjustments depending on the molecular context. Glucose oxidation changes during breast cancer progression since less aggressive tumors, such as luminal A and luminal B, show increased oxidative phosphorylation and decreased extracellular acidification, while the opposite has been found in more aggressive tumors, such as triple-negative breast cancer (TNBC) [35]. Indeed, several glycolytic enzymes are deregulated in human breast cancer, including PKM2 (pyruvate kinase muscle isozyme 2) [37], PFK (phosphofructokinase) [38,39], and HK2 (hexokinase 2) [40,41], and the pharmacological inhibition of HK2 has been investigated as a therapeutic strategy [42]. Supporting the glycolytic phenotype in more aggressive tumors with poor prognosis, higher expressions of lactate dehydrogenase B (LDHB) and lactate transporters (MCT1 and MCT4) [43,44] and the inhibition of lactate secretion into the microenvironment to suppress tumor growth have been reported [45]. Therefore, this body of evidence shows that the critical point of glucose metabolism is the key enzyme that regulates glycolysis functioning.

Lipids and amino acid metabolism are also dysregulated. Studies have reported that the amino acid profile either in tumors or blood is distinct from the healthy condition [46,47,48]. Glutamine is the most abundant amino acid in the bloodstream and serves not only as a source of carbon for TCA cycle intermediates, but also to reduced coenzymes formation such as NADH and FADH_2_. In turn, these fuel oxidative phosphorylation in the electron transport chain (ETC). TCA cycle intermediates from glutamine are generated via glutaminolysis, performed mainly by glutaminase 1 (GLS) and glutaminase 2 (GLS2) [49,50,51]. Both glutaminases convert glutamine into glutamate via reductive carboxylation, which enters the TCA as α-ketoglutarate [49]. In breast cancer, GLS and GLS2 are overexpressed at distinct levels depending on the histological subtype, and cells may display dependency on glutamine to survive [51], GLS2 being investigated as a protumorigenic gene [23]. This scenario served as inspiration for clinical trials with glutaminase inhibitors [52,53,54], but further studies are required. Carbon from glutamine may be channeled to fatty acid synthesis and further processed into phospholipids for biological membranes. Lipid synthesis may occur endogenously from either glutamine or glucose, a biological process named de novo lipogenesis (DNL), which is performed by FASN and Acyl-CoA carboxylase (ACC). In TNBC, FASN inhibition has an antitumor effect even at low protein levels of expression [55,56], and DNL has been reported as required for brain metastasis in humans [27]. 

The present scenario, built mostly with findings regarding human breast cancer, demonstrates that metabolic reprogramming could be a fragility, therefore becoming a therapeutic opportunity. Moreover, the literature reports several key points in mammary gland cell metabolism, showing that a large number of pathways, both catabolic and anabolic, are deregulated. However, such alterations are poorly explored in the canine mammary glands and leave an information gap in the veterinary community (Figure 1). Therefore, this review aimed to put together “*state-of-the-art*” metabolic adjustments described in CMTs that could be a target in cancer treatment and prevention.

## 3. Metabolic Reprogramming in Canine Mammary Tumors

### 3.1. Carbohydrate Metabolism

Glucose metabolism is among the most frequently altered metabolic pathways in human breast cancer [34,35,57]. Although information regarding female dogs is not deeply understood, the same seems to occur in CMTs. Increased blood glucose levels have been reported in dogs bearing mammary neoplasms, and this seems to be related to the presence of tumor cells given that, upon mastectomy, the levels drop considerably [58]. This has been linked to the release of higher levels of lactate by cancer cells and its further conversion into glucose through gluconeogenesis in the liver. Jayasri and colleagues (2016) [59] reported on the deregulation of glycolysis in mammary tumors, supporting the increase in glucose uptake and processing, as well as the channeling of its carbons to biosynthetic pathways. In CMTs, the authors found an increase in hexose levels of 1.75-fold and 1.6-fold for HK. The expression of pyruvate kinase 2 (PKM2), the last enzyme of the glycolytic cascade, was also overexpressed in malignant tissue, its levels being correlated with the tumor grade [60]. These findings corroborate previous data that reported increased expression in glycolytic enzymes [61] and also supported the increase in GLUT1 expression observed in complex mammary carcinoma [62]. Interestingly, higher GLUT1 and GLUT3 expression can be detected under hypoxia, which is often observed in large tumors [63]. However, in tumors with normal expressions of TP53, glycolysis in canine mammary carcinomas seems to respond differently because phosphoglycerate mutase 2 (PGAM1), an enzyme that converts 3-phosphoglycerate into 2-phosphoglycerate, is downregulated [64]. The role of glucose metabolism has been found to not only be limited to energy supply for tumor growth but also a signaling pathway. The higher glycolysis rate in CMT cells increases the release of lactate, which, in turn, polarizes macrophages into the M1/M2 intermediate state [65]. M2-like macrophages can be activated by tumor-derived lactate and exert their anti-inflammatory function, promoting immune escape, which is correlated with tumor progression and aggressiveness [66,67]. However, the function of the microenvironment, mainly immune cells, in CMTs is poorly explored, though in vitro study models have been suggested [68]. In addition, not only are glucose breakdown-related metabolites altered but also those related to derivative pathways. Decreased glucose-6-phosphatase and fructose-1,6-bisphosphatase have been found to be decreased in CMT [59], both related to gluconeogenesis. Also, TLK and TLK-1 have been shown to be differentially expressed, the former being highly expressed in benign and malignant tumors compared with normal tissue, while TLK-1 levels are higher mainly in hyperplastic lesions, simple adenomas, and simple carcinomas [69]. This indicates that carbohydrate catabolism and anabolism are essential to meeting the demands of tumor growth. Indeed, PPP enzymes have been shown to be deregulated in CMT, but the estrogen context seems to play an additional role because cells positive for estrogen receptor expression have low expressions of glucose-6-phosphate dehydrogenase (G6PDH), whereas those that do not express estrogen receptors have higher levels [70].

Glucose metabolism reprogramming is a hallmark of human carcinogenesis and offers a tool for diagnosis, including in the veterinary approach. Since tumors usually uptake higher levels of glucose, ^18^F-FDG-PET/CT has been adopted for imaging. Sánchez and colleagues (2019) [71] showed that such a technique is useful in distinguishing malignant from benign tumors but not for histologic subtypes or grades. Moreover, given that glucose uptake is increased, metformin, a drug used to control glucose levels in diabetes, has been tested as an antiproliferative agent in vivo and in vitro in the context of canine mammary carcinomas [72]. Together, this evidence indicates that glucose is the core of carbohydrate metabolism and is sharply related to tumor initiation and progression, but it is still poorly explored in CMTs, mainly the underlying mechanisms. In addition, a better understanding of carbohydrate metabolism may improve the diagnosis and offer a more personalized treatment. 

### 3.2. Lipid Metabolism

Obesity in dogs is an epidemiological issue that has been widely investigated given its associated diseases, such as inflammation, neoplasia, and cardiovascular disease [73]. Overweight or obese female dogs have been reported with earlier-onset CMTs and higher histologic grades compared with those that are either lean or have optimal body weight [74]. Moreover, obese dogs are associated with more aggressive CMTs, angiogenesis, and tumor-associated macrophage (TAM) infiltration, this being considered a risk factor for the incidence and progression of mammary neoplasms [75,76]. Adiponectin levels, usually observed under obesity conditions, were found to be decreased in overweight or obese individuals, while the number of macrophages increased, as noted by the authors of [77]. These alterations correlate with poor prognosis when a high histological grade and lymphatic invasion are found [77]. The proportion of grade I tumors is higher among leptin-positive CMTs than leptin-negative CMTs, and a positive correlation has been observed with progesterone receptor-positive tumors. Also, the number of tumors with positive estrogen receptor expression is higher in CMTs with leptin receptor (ObR) expression than those without it. The aforementioned authors suggested that increased adiponectin expression may prevent cancer development and positively affect the prognosis of CMTs, whereas decreased expression in obese dogs influences their aggressiveness [77]. Moreover, the same authors, in another paper, found that the increased expression of aromatase is correlated with hormone receptor-positive tumors [74]. The leptin receptor and aromatase both increase with obesity, but their association with the incidence of CMTs remains up for debate. A case–control study based on interviews with owners found no correlation between high-fat diets or obesity and CMT occurrence in female dogs [78], while others have shown that canines fed a high-fat diet have a higher survival rate compared with those with a low-fat diet [79]. 

In addition to obesity inducing several metabolic alterations, disparities in the literature may be due to the fact that fatty acids acquired through diet do not affect tumor metabolism via quantity alone but also due to their quality. In rapidly proliferating cells, lipids are required for membrane synthesis, energy supply, cell signaling, and post-translational modifications. The carnitine shuttle system is a key player in lipid catabolism since it allows for the translocation of fatty acids from the cytoplasm into the mitochondria for further oxidation and ATP production. Carnitine acylcarnitine translocase (CACT), carnitine palmitoyl transferase 2 (CPT2), and carnitine O-acetyltransferase (CrAT) have been found to be overexpressed in CMTs in cell lines and tissue compared with normal conditions, except for decreases observed in poorly undifferentiated, higher-grade CMTs [80]. In the same manner, carnitine palmitoyl transferase 1 A (CPT1A), a rate-limiting enzyme of fatty acid oxidation located in the outer mitochondrial membrane, has been reported as overexpressed in differentiated CMTs compared with normal tissue both in vivo and in vitro, whereas a decrease in CPT1A expression has been observed in less differentiated tumors [81]. These findings suggest that lipid metabolism is rewired along carcinogenesis, given that cells seem to rely on energy derived from lipids, but they probably shift to a glycolytic phenotype. 

In different experimental models, as well as in clinical data, saturated fatty acids (SFAs) have been shown to drive carcinogenesis because of the MYC program in several human cancers [82], which can also be observed in breast cancer [83,84]. Interestingly, in spheroids from canine mammary adenocarcinoma, the levels of palmitoleate, palmitate, and dihomo-gamma-linolenic acid are higher compared with adherent cells, suggesting that certain SFAs are required for tumor formation [85]. Polyunsaturated fatty acids (PUFAs) have been investigated because of their antitumoral properties in different cancers [31,32,86,87,88,89]. Conjugated linoleic acid (CLA), one of the most abundant PUFAs available in the diet, decreases the growth of epithelial and stromal CMT cells through the suppression of COX-2 and the prostaglandin E_2_ receptor (EP2) [90]. This is a promising outcome since CMT cells may express COX-2 and produce high levels of PGE_2_ [91]. However, the CLA antiproliferative effect seems to depend on its configuration, given that trans-10,cis-12 increases the expression of cell-cycle-progression-related genes and cis-9,trans-11 stimulates apoptotic genes in CMT explants [92]. In human breast cancer, PUFAs from the omega-3 class have been shown to increase patient survival and serve as a preventive agent [93], although this is still under debate [94]. In companion animals, fish oil supplementation, the main source of omega-3 PUFAs, has been recommended for several conditions, such as renal disease, cardiac and skin inflammatory disorders, dyslipidemia, and cancer [95]. Despite assessed on a very limited simple size, Tuzlu and colleagues (2021) [96] reported that PUFAs belonging to the omega-3 class were higher in healthy dogs compared with those bearing CMT, while an opposite correlation was observed for omega-6 fatty acids, suggesting a protective property in the former. In humans, at the mechanistic level, the omega-3 docosahexaenoic acid (DHA) was able to induce cell death in MDA-MB-231 cells [97], but also cell cycle modulation in other cancers, such as prostate cancer [32]. In dogs, a clinical trial using long-chain omega-3 fatty acid supplementation through fish oil (eicosapentaenoic acid, EPA; 29 g/kg of diet and DHA 24 g/kg of diet) showed that increased DHA content is associated with longer disease-free intervals and survival in dogs with stage III lymphoma [98]. In TRAMP mice, a DHA-rich diet was able to delay prostate cancer progression [31]. Therefore, this body of evidence suggests the potential antitumoral effects of omega-3 in canine tumors, though its specific role in CMT remains elusive. 

Taken together, this evidence indicates that either endogenously synthesized fatty acids or exogenous lipids obtained through diet may affect tumor growth. Also, despite being widely studied in humans, lipid metabolism is poorly investigated in CMTs, but it could be a new venue as a preventive and therapeutic strategy, as it has been for human breast cancer. 

### 3.3. Amino Acids Metabolism

Amino acids and their transporters play a plethora of roles in cells, including protein synthesis, oxidative status, the regulation of protein conformation, excretion through the urea cycle, and cell signaling, but this study will limit their role to cell metabolism, acting directly or indirectly. Plasma-free amino acid (PFAA) levels were investigated in dogs bearing or not bearing CMTs and with or without metastasis in [99]. Compared with healthy animals, methionine, serine, asparagine, glutamine, alanine, taurine, and citrulline plasma levels decreased in the CMT group. In the metastatic group, methionine, lysine, histidine, aspartate, serine, asparagine, glutamate, glutamine, alanine, taurine, citrulline, and ornithine plasma levels increased compared with healthy dogs [99]. This evidence suggests that distinct amino acids may play a role in CMT initiation and progression and during metastasis. Alterations in amino acid profiles due to the presence of tumors have also been observed in mice bearing xenograft tumors of human MDA-MB-231 triple-negative cells, resulting in the deregulation of arginine and proline metabolism, the urea cycle, and aspartate metabolism [100]. Moreover, the amino acid profile in plasma varies depending on the light cycle, showing *cross-talk* between the circadian rhythm and amino acid metabolism in breast cancer [100]. It would be reasonable to consider such associations in CMTs given their similarities with human breast cancer, but further investigation is required to elucidate this issue.

In human TNBC, an aggressive tumor subtype with a poor prognosis [101,102], cells are dependent on glutamine metabolism, and studies have shown that the inhibition of glutaminase A (GAC) leads to reductions in survival and cell proliferation [103]. GAC is often overexpressed in human breast cancer, and it has been demonstrated via immunohistochemistry and protein expression that GLS follows the same pattern in CMTs, as it is highly expressed in high-grade tumors [104]. Although further studies are required to elucidate this issue, this is a promising target since glutaminase inhibitors such as CB-839 have been proposed as therapeutic opportunities [103]. Transglutaminase II (TGase II) has also been reported as overexpressed in canine tumors, including in mammary tumors, and this was linked to survival mechanisms [105]. The expression of amino acid transporters has also been reported to be altered in CMTs. L-type amino acid transporter 1 (LAT1) is related to the uptake of branched or aromatic amino acids, such as leucine, isoleucine, valine, phenylalanine, tyrosine, tryptophan, methionine, and histidine, in a sodium-independent manner [106]. While upregulated in CMT, it detains low expression in normal tissue, being LAT1 inhibition a potential therapeutic strategy [106,107]. In addition, it is speculated that citrulline, a non-essential and non-proteinogenic amino acid involved in the urea cycle, also plays a role in CMT since peptidylarginine deiminase 2 (PAD2), an enzyme that converts arginine into citrulline, decreases in CMTs compared with normal tissue [108]. In this context, PAD2 was shown to be responsive to the epidermal growth factor (EGF), but not estrogen or progesterone, in the canine mammary primary carcinoma cell line CMT25 [109]. Taken together, this evidence shows that the amino acid profile and metabolism seem to be altered in CMTs.

### 3.4. Mitochondrial Metabolism

Mitochondria is known as the powerhouse of the cell because of its ATP production property, which meets bioenergetic needs. According to the revised Warburg effect, its activity is mostly channeled to biosynthetic pathways and energy production would not be its main role. In CMTs, the most frequent mitochondrial alteration reported in the literature is to its DNA (mtDNA), which suggests organelle plasticity to support carcinogenesis. It has 16,727 pb as a reference sequence, encoding two ribosomal RNA, twenty-two transfer RNA, and thirteen polypeptides related to the ETC [110,111]. The high heterogeneity of mtDNA can be observed in distinct CMTs [112] and might be explained by a lack of histones and repair mechanisms, in addition to mitochondria being the main source of ROS. Oxidative stress is related to carcinogenesis, and a number of different markers have been identified in CMTs, such as increases in lipid oxidation and the altered activities of antioxidant enzymes [113,114,115]. Studies have demonstrated that defects in mtDNA are potential risk factors for CMT. Surdyka and Slaska [116] identified a sequence of 26 polymorphic loci and five mutations, revealing that the mitochondrial displacement loop is a hotspot for mutation in CMT, with these alterations being correlated with dog size. Also, Surdyka and Slaska [117] reported that ND2 (NADH dehydrogenase subunit 2), COXII (cytochrome c oxidase subunit II), ATP6 (ATP synthase F0 subunit6), and COX3 (cytochrome c oxidase subunit III), most of them related to the ETC function, are mutated in CMT, which is a hotspot too. Moreover, such alterations have been found in both tumors and blood, with ATP6 and COXII exclusively in the latter case [117]. This is of particular interest because it reveals heteroplasmy and also that specific mitochondrial mutations might be required for niche formations along with metastasis. Surprisingly, such variability was found in the same dog bearing two tumors in [112]. Despite this evidence allowing us to infer alterations in mitochondrial metabolism, no study has analyzed mitochondrial function, especially among cancer subtypes. However, as aforementioned, CPT1A is overexpressed in CMT [81], suggesting a higher mitochondrial activity compared with normal conditions, but further studies are required to address this issue.

A high number of alterations in mitochondria indicates their crucial role in canine mammary carcinogenesis, which may also be a vulnerability. In addition to metabolism, mitochondria are closely related to cell death, which has been explored in vitro as a therapeutic strategy for CMT [118], including by our research group [119]. The administration of iodine (I_2_) combined with doxorubicin in dogs with mammary tumors has been shown to improve the therapeutic outcomes associated with mitochondrial membrane oxidation [120]. Moreover, induced mitochondrial dysfunction has an antitumor effect on CMT cells from metastasis [121]. 

## 4. Conclusions

Metabolic reprogramming in CMTs is summarized in Figure 2 and Table 1, showcasing the most commonly described alterations compared with healthy individuals, as well as their sources, in vivo or in vitro. However, studying metabolic rewiring in CMTs is challenging given the limited number of samples, their high variability, the lack of well-characterized tumor subtypes, and the absence of suitable in vitro and in vivo experimental models. These factors have led to descriptive studies and potential data misinterpretations, underscoring the need for new investigations that focus on understanding the underlying mechanisms and employing improved experimental designs. Several metabolic alterations observed in CMTs are also described in human breast cancer, particularly in glucose, amino acid, and lipid metabolism. However, despite numerous clinical trials attempting to target metabolic vulnerabilities in humans, most of them have not been tested on dogs or have very limited reported information. Tumor metabolism has garnered attention in the past decade because of its potential as the “*Achilles heel*” of neoplastic cells, and its complexity has increased with its interaction with the microenvironment. It has been reported that tumor cells can secrete metabolites that regulate immune cell populations, such as macrophages. The polarization of macrophages, particularly the M2 type, plays a crucial role in suppressing inflammation, which can favor immune evasion, cancer progression, and metastasis. However, studies examining the association between these two variables in CMTs are scarce, despite the potential to improve the outcomes of current therapies. Collectively, this body of evidence highlights new avenues for therapeutic approaches to canine mammary cancer, uncovering a plethora of metabolic vulnerabilities that should be explored. By delving into these vulnerabilities, we can potentially enhance the disease-free interval and overall survival of affected animals.

## Figures and Tables

**Figure 1 animals-13-02757-f001:**
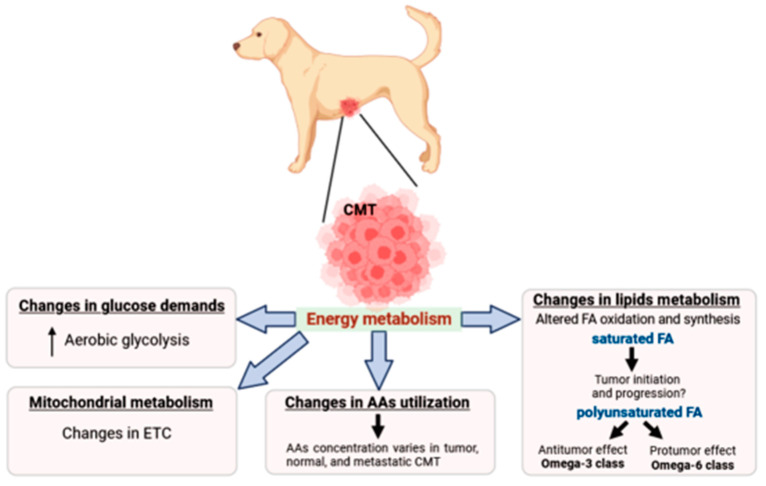
*Alterations in canine mammary tumor energy metabolism.* It has been frequently observed that metabolic rewiring in female dogs involves changes in glucose demands (Warburg effect), mitochondrial metabolism, amino acid profiles, and lipid metabolism. Tumors commonly increase glucose uptake to meet their anabolic demands, with energy metabolism adjusted for amino acids and fatty acid usage being adjusted as well. The role of distinct fatty acids regarding their unsaturation status is not clear and requires further investigation. Legend: CMT—canine mammary tumor; AAs—amino acids; FA—fatty acid; ETC—electron transport chain.

**Figure 2 animals-13-02757-f002:**
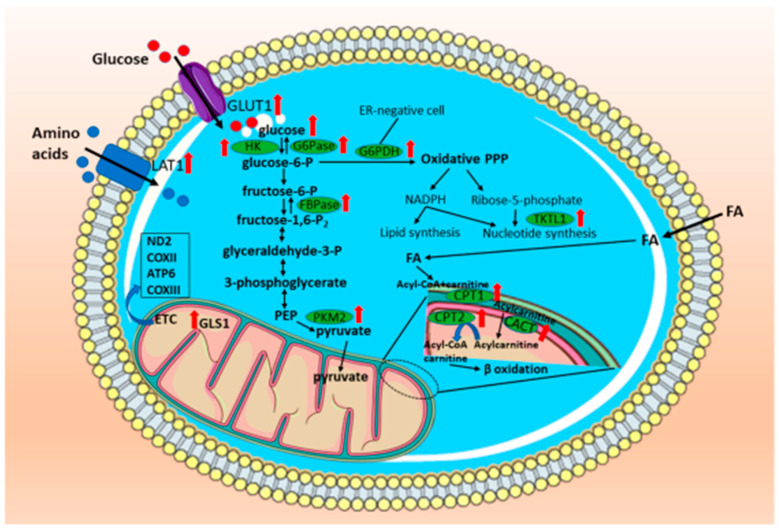
*Summary of metabolic alterations in canine mammary tumors*. Changes in carbohydrate, amino acid, lipid, and mitochondrial metabolism were found. Glucose uptake and glycolysis enzymes HK and PKM2 increase in CMTs. This pattern is also observed for gluconeogenesis and the pentose phosphate pathway. Amino acid utilization has been reported to be altered, especially in glutamine metabolism via glutaminase 1 and amino acid transport. In addition, fatty acid oxidation increases because of the overexpression of enzymes related to the carnitine system. Such alterations involve mitochondrial metabolism, which is affected in CMTs mainly by ETC genes, given that they are described as a hotspot for mutations in mtDNA. Legend: GLUT1—glucose transporter 1; HK—hexokinase; G6Pase—glucose-6-phosphatase; FBPase—fructose-1,6-bisphosphatase; PKM2—pyruvate kinase muscle 2; LAT1—L-type amino acid transporter 1; TKTL1—transketolase-like 1; CPT1—carnitine palmitoyl transferase 1; CPT2—carnitine palmitoyl transferase 2; CACT—carnitine acylcarnitine translocase; ND2—NADH dehydrogenase subunit 2; COXII—cytochrome c oxidase subunit II; ATP6—ATP synthase F0 subunit 6; COXIII—cytochrome c oxidase subunit III; ER—estrogen receptor; FA—fatty acid; GLS1—glutaminase 1; PPP—pentose phosphate pathway; ETC—electron transport chain; mtDNA—mitochondrial DNA.

**Table 1 animals-13-02757-t001:** *Metabolic alterations in canine mammary tumors sorted by source and metabolic pathways*. Findings on CMTs available in the literature were almost entirely collected from tissue or blood and minimally obtained from cell cultures. Legend: GLUT1—glucose transporter 1; HK—hexokinase; PKM2—pyruvate kinase muscle 2; LAT1—L-type amino acid trans-porter 1; TLK-1—transketolase-like 1; CPT1—carnitine palmitoyl transferase 1; CPT2—carnitine palmitoyl transferase 2; CACT—carnitine acylcarnitine translocase; ND2—NADH dehydrogenase subunit 2; COXII—cytochrome c oxidase subunit II; ATP6—ATP synthase F0 subunit 6; COXIII—cytochrome c oxidase subunit III; GLS1—glutaminase 1; mtDNA—mitochondrial DNA; CLA—conjugated linoleic acid; DHA—docosahexaenoic acid; PAD2—peptidylarginine deiminase 2; Met—methionine; Ser—serine; Asn—asparagine; Gln—glutamine; Ala—alanine; Tau—taurine; Cit—citrulline; Orn—ornithine; G6Pase—glucose-6-phosphatase; FBPase—fructose-1,6-bisphosphatase.

Metabolic Pathway	Alteration in CMTs	Source	Reference
** Glycolysis **	Glucose levels	in vivo	Rodigheri et al., 2023 [58]; Jayasri et al., 2016 [59]
	Glycolysis enzymes	in vivo	Jayasri et al., 2016 [59]; Arai et al., 1997 [61]
	HK	in vivo	Jayasri et al., 2016 [59]
	PKM2	in vivo	Lee et al., 2020 [60]
	GLUT1 and GLUT3	in vivo	Freeman et al., 2010 [62]; Mees et al., 2011 [63]
	TLK and TLK-1	in vivo	Burrai et al., 2017 [69]
** Pentose phosphate pathway **	G6PDH	in vivo	Nerurkar et al., 1990 [70]
** Gluconeogenesis **	G6Pase and FBPase	in vivo	Jayasri et al., 2016 [59]
** Fatty acid oxidation **	CACT, CPT2, CrAT, CPT1A	in vivo and in vitro	Cacciola et al., 2021 [80]; 2020 [81]
** Omega-3 fatty acids **	CLA	in vitro, primary cell culture	Wang et al., 2006 [90]
	DHA	in vivo	Tuzlu et al., 2021 [96]
** Amino acids **	Met, Ser, Asn, Gln, Ala, Tau, Cit, and Orn	in vivo	Azuma et al., 2012 [99]
** Amino acid-related enzymes **	TGase II	in vivo	Wakshlag et al., 2006 [105]
	GLS1	in vivo	Ryu et al., 2018 [104]
	LAT1	in vitro	Fukumoto et al., 2013a [106], 2013b [107]
	PAD2	in vivo	Cherrington et al., 2012 [109]
** Mitochondria **	mtDNA	in vivo	Bertagnolli et al., 2009 [111]
	ND2, COXII, ATP6, COX III	in vivo	Surdyka et al., 2017 [117]

## Data Availability

Data sharing is not applicable.

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
