# Peer review of "Metabolic Alterations in Canine Mammary Tumors"

_animals, 2023, doi:10.3390/ani13172757_

Round 1

Reviewer 1 Report

This review makes a valiant and worthwhile effort to collate evidence for metabolic reprogramming in canine mammary cancer.  The data is used to support the argument that various metabolic derangements may present opportunities for therapeutic targeting.   However the paper is difficult to read and needs probably a couple more drafts to make it more concise and polish the English, and to rationalise the structure.  The paper needs moderate to extensive attentive editing and some suggestions to improve it follow:

1. Readability - many sentences make no sense e.g. Line 37 sentence starting "Several elements contribute ..."  This entire sentence could be deleted as it adds no useful information to the manuscript.  There are many other examples through the paper.  If a sentence has no specific point, remove it.

2. Correct referencing needs to be carefully checked e.g. Line 55 references 13 and 14 do not support what the sentence claims.  Lines 160-162 : this sentence misinterprets reference 66.   These and other examples indicate that all the referencing needs careful verification; references can easily get mixed up during early drafts of a paper.  

3. The authors need to distinguish between benign and malignant mammary tumours.   Blanket statements should be avoided unless qualified, for example line 60 "...in most cases with poor prognosis..." is true for some but not most mammary tumours in dogs.

4. Author list:  ..., Zuccari and D.A.P.C.  Remove "and" 

5. I would like to see a concise comparison of the clinicopathologic and incidence rate similarities between human and canine mammary malignancies (I assume you're not interested in the benign tumours ... or are you?).

6. Line 219 what is ObR?

7. Unless absolutely required, remove information about non-mammary cancers - it is confusing and makes the paper more difficult to read.

8. Spell out acronyms the first time you use them e.g. TNBC

9. Overall structure needs review - Section 2 seems to be a fairly concise summary of human mammary cancer metabolic information, Section 3 is about metabolic reprogramming in dogs and seems to be divided into several subsections but the naming is inconsistent. Section 3.1 is about carbohydrates Should Section 4 be renamed Section 3.2 Lipids, Section 5 should be Section 3.3 amino acids etc.   Information in these sections is mixed up between humans and dogs : I would leave the dog information in place and delete the human information or summarise it back in Section 2 if it has not already been done.

10. I noticed very little critical appraisal of the papers cited, at least some of which are based on poorly conducted or overinterpreted clinical investigations.

11. Another point about referencing - refer to original work rather than other reviews e.g. Gray et al is not a good reference for the first sentence in the introduction  - you should really find where that paper sourced information 

12. at the outset in this manuscript, you should make it clear that when you state, for example, in line 1 "the most common cancer in female dogs" you mean intact female dogs that have not been surgically neutered.

13. Conclusion - there are a lot of potential targets - what do you think are the two or three most promising for future investigation? 

See Comments & Suggestions for Authors. 

Multiple singular/plural errors, past tense/present tense errors, the/a errors, verbose rather than concise, syntax mixups.   Most of these can be fixed with closer attention to the text.

Author Response

Editorial Commission of Animals – Special issue ‘Canine Mammary Tumors

We are grateful to the referees for their valuable comments and suggestions. The present version of the manuscript has been altered in the light of the comments, as detailed below. All changes made to the manuscript and the revised version is being presented according to the journal’s rule. Changed text appears highlighted. Also, we responded to all raised issues.

            We take the opportunity to thank the reviewers for all their valuable comments and suggestions to improve the quality of our manuscript.

Sincerely yours,

Profa. Dra. Debora A.P.C. Zuccari

Laboratory of Molecular Investigation in Cancer/LIMC

Faculdade de Medicina de São José do Rio Preto/FAMERP

São José do Rio Preto, SP, Brasil.

Response to reviewers:

  1. Readability - many sentences make no sense e.g. Line 37 sentence starting "Several elements contribute ..." This entire sentence could be deleted as it adds no useful information to the manuscript. There are many other examples through the paper.  If a sentence has no specific point, remove it.

Reply: We appreciate the reviewers’ suggestions concerning the text editing. Sentence in Line 37 was removed in addition to others that looked pointless to the authors.

  1. Correct referencing needs to be carefully checked e.g. Line 55 references 13 and 14 do not support what the sentence claims. Lines 160-162 : this sentence misinterprets reference 66. These and other examples indicate that all the referencing needs careful verification; references can easily get mixed up during early drafts of a paper. 

Reply: Referencing was checked and properly updated.

  1. The authors need to distinguish between benign and malignant mammary tumours. Blanket statements should be avoided unless qualified, for example line 60 "...in most cases with poor prognosis..." is true for some but not most mammary tumours in dogs.

Reply: Benign tumors were considered as a mass limited that grow slowly, it is limited to its site of occurrence with low invasive and metastatic capacity. Malignant tumors usually grow faster and invade the surrounding tissues with high capacity to metastasize. In the present review, canine mammary tumors were restricted to malignant phenotype, unless otherwise stated. This issue was clarified in Lines 51-52.

  1. Author list: ..., Zuccari and D.A.P.C. Remove "and"

Reply: Author list was corrected. Indeed, the word “and” was not inserted in the main text submitted for review, but edited along text processing by MDPI assistants.

  1. I would like to see a concise comparison of the clinicopathologic and incidence rate similarities between human and canine mammary malignancies (I assume you're not interested in the benign tumours ... or are you?).

Reply: We appreciate the reviewer suggestion and agree that such comparison would improve our manuscript. However, Editors and other reviewer both requested to remove any information regarding human breast cancer, emphasizing only canine mammary tumors. Therefore, we decided to follow their requirements for submission.

  1. Line 219 what is ObR?

Reply: ObR stands for leptin receptor. This information was inserted in the main text Line 218.

  1. Unless absolutely required, remove information about non-mammary cancers - it is confusing and makes the paper more difficult to read.

Reply: Information regarding non-mammary tumors was revised and removed as suggested.

  1. Spell out acronyms the first time you use them e.g. TNBC

Reply: Acronyms were properly addressed.

  1. Overall structure needs review - Section 2 seems to be a fairly concise summary of human mammary cancer metabolic information, Section 3 is about metabolic reprogramming in dogs and seems to be divided into several subsections but the naming is inconsistent. Section 3.1 is about carbohydrates Should Section 4 be renamed Section 3.2 Lipids, Section 5 should be Section 3.3 amino acids etc. Information in these sections is mixed up between humans and dogs : I would leave the dog information in place and delete the human information or summarise it back in Section 2 if it has not already been done.

Reply: Thank you for this suggestion. The text file uploaded during submission did not contain any Sections, just subtitles. The section numbering was performed by the MDPI assistant, and the revised text was corrected after your comment. Regarding information on human and dogs, we were requested by Editors to remove human data and leave only canine data in the manuscript. This generated the Section 2 which is an overview of the metabolic reprogramming in general, even though most of the citations refers to breast cancer because most of information in the literature was collected in humans. However, along the text you may eventually find information out of Section 2 about human breast cancer that was inserted to highlight either specific similarities or differences between species with an objective. However, we appreciate the reviewer’s suggestions and, unless necessary, human information was changed to Section 2 or removed.

  1. I noticed very little critical appraisal of the papers cited, at least some of which are based on poorly conducted or overinterpreted clinical investigations.

Reply: Thank you for your comment. To the best of our knowledge, this is the first review of its kind and was conceptualized after notice that there is a huge lack of information regarding the metabolic alteration in canine mammary tumors compared to human breast cancer. Also, that few available information was mostly descriptive without an elucidation of the underlying mechanism, or the sample size was very limited. Indeed, this could lead to misinterpretation, but our aim was to reunite the published information, open a debate about the issue and raise questions that were not fully addressed or remain elusive. This limitation was highlighted in the Conclusion Section. In addition, comments were added in the statements to highlight study limitations, as seen in Lines 360-380.

  1. Another point about referencing - refer to original work rather than other reviews e.g. Gray et al is not a good reference for the first sentence in the introduction - you should really find where that paper sourced information

Reply: References were checked, and reviews avoided unless absolutely convenient to the topic.

  1. at the outset in this manuscript, you should make it clear that when you state, for example, in line 1 "the most common cancer in female dogs" you mean intact female dogs that have not been surgically neutered.

Reply: Thank you for your suggestion. Indeed, we referred to intact female dogs. This alteration was done as requested in Line 36.

  1. Conclusion - there are a lot of potential targets - what do you think are the two or three most promising for future investigation?

Reply: Thanks for your comment. It is hard to define what would be the promising targets, but up to date glutaminase and fatty acid synthase inhibitors have received efforts from researchers in human breast cancer. Since there are several metabolic alterations shared between humans and dogs, these could be new opportunities in canine mammary tumors as well. This consideration was inserted in the main text.

Reviewer 2 Report

The manuscript is a review of the metabolic changes associated with CMTs. Although the theme is no doubt interesting, the present manuscript presents some flaws that hinder its proper appreciation.

It presents a considerable number of references, but the authors do not mention the criteria used for research and article selection. Some recent contributions on the subject were not included in this review, such as:

Razzuoli, E., De Ciucis, C. G., Chirullo, B., Varello, K., Zoccola, R., Guardone, L., ... & Modesto, P. (2022). Molecular Characterization of CF33 Canine Cell Line and Evaluation of Its Ability to Respond against Infective Stressors in Sight of Anticancer Approaches. Veterinary Sciences, 9(10), 543.

Maués, T., Oliveira, T. F. D., El-Jaick, K. B., Figueiredo, A. M. S., Ferreira, M. D. L. G., & Ferreira, A. M. R. (2021). PGAM1 and TP53 mRNA levels in canine mammary carcinomas–Short communication. Acta Veterinaria Hungarica, 69(1), 50-54.

The text also cites studies on canine tumours other than mammary tumours, without clearly referring so. An example is the sentence in lines 198 and 199, including a research article on hemangiosarcoma along with the other two, both including results on canine mammary tumours. The same lack of precision is present in the manuscript concerning species; often it is not clear whether the authors are referring to human cancer or canine cancer such as in line 270, when mentioning prostate (cancer?). Considering that this is presented as a review on the metabolic changes present in CMTs, it would be expected that whenever resorting to refer studies on other types of tumours and species other than the canine, this should always be clarified.

 A more serious flaw is the lack of precision when using terms such as “stage” (see lines 187 and 188). In Oncology, stage is not a synonym for malignancy or benignity and it should not be used as such. Cancer staging is different, and it is not to be confused with grade, either. Likewise, the term “subtype” seems to be used indifferently to refer to different histological subtypes, to malignant vs. benign tumours (see line 198, since the cited references describe how 18F-FDG-PET/CT may be useful to distinguish malignant and benign tumours, to detect metastasis and for tumour staging but it is inaccurate to consider it distinguishes histological subtypes). In line 355 it is said “high heterogenicity” is observed in “all CMT subtypes”, although the cited work by Kowal et al. is only on malignant mammary tumours (and not on benign CMTs, which have several histological subtypes too).

 Overall,how this work is written lacks clarity and English should be extensively edited.

Sometimes a phrase begins, and it is not clear what the subject is. An example of this (lines 41 to 45):

“Sexual hormones deregulation, such as exposure to endogenous ovarian hormones, may cause the mammary tumors development in dogs [3–5], but others may also influence CMT such as obesity in early stages of life [6], inflammation [7] and the increase in free radicals and reactive oxygen species (ROS) [8]. They may appear as single or multiple nodules, and posterior…”

 To whom does the plural subject “They” refer? To sexual hormones? To ROS? There are multiple similar phrases throughout the text. In l,nes 162-163 it is said “considerably [66]. This has been linked to higher lactate production and release by cancer cells, a well-known feature of the Warburg effect, which is converted to glucose through gluconeogenesis in the liver.”. As it is, the sentence suggests that the Warburg effect is converted to glucose through gluconeogenesis by the liver.

There are also many long sentences, lacking punctuation and some phrases that seem unfinished (see, for example, lines 303 and 304) or that need to be re-written (see lines 324 and 325).

The term “hormone positive cells” could probably be replaced by “hormone receptor positive cells”.

TNBC is an abbreviature, but it is never explained in the text to what it refers to.

Some conclusions may be considered quite bold such as when it is said “Therefore, this evidence shows that, from the metabolic standpoint, the amino acid profile is altered in CMT and that glutamine metabolism seems to follow the same response in dogs and humans. “, as the authors on the following sentence write “ Although further studies are required to elucidate this issue, this is an important finding (…)”. If it is well described that the amino acid profile is altered in CMTs, saying there is evidence that glutamine metabolism follows the same response in dogs and humans when CMTs are yet to be fully characterized concerning this aspect may be a bit forceful.

Due to these issues, I believe the manuscript should be extensively edited and corrected before considering its publication.

The artwork is adequate but the addition of a table summarizing the findings exclusively in canine mammary tumours, both in vivo and in vitro, with cited sources, would be a welcome addition.

The manuscript needs extensive editing. 

Author Response

Editorial Commission of Animals – Special issue ‘Canine Mammary Tumors’

We are grateful to the referees for their valuable comments and suggestions. The present version of the manuscript has been altered in the light of the comments, as detailed below. All changes made to the manuscript and the revised version is being presented according to the journal’s rule. Changed text appears highlighted. Also, we responded to all raised issues.

            We take the opportunity to thank the reviewers for all valuable comments and suggestions to improve the quality of our manuscript.

Sincerely yours,

Profa. Dra. Debora A.P.C. Zuccari

Laboratory of Molecular Investigation in Cancer/LIMC

Faculdade de Medicina de São José do Rio Preto/FAMERP

São José do Rio Preto, SP, Brasil.

Response to reviewers:

  1. Readability - many sentences make no sense e.g. Line 37 sentence starting "Several elements contribute ..." This entire sentence could be deleted as it adds no useful information to the manuscript. There are many other examples through the paper.  If a sentence has no specific point, remove it.

Reply: We appreciate the reviewers’ suggestions concerning the text editing. Sentence in Line 37 was removed in addition to others that looked pointless to the authors.

  1. Correct referencing needs to be carefully checked e.g. Line 55 references 13 and 14 do not support what the sentence claims. Lines 160-162 : this sentence misinterprets reference 66. These and other examples indicate that all the referencing needs careful verification; references can easily get mixed up during early drafts of a paper. 

Reply: Referencing was checked and properly updated.

  1. The authors need to distinguish between benign and malignant mammary tumours. Blanket statements should be avoided unless qualified, for example line 60 "...in most cases with poor prognosis..." is true for some but not most mammary tumours in dogs.

Reply: Benign tumors were considered as a mass limited that grow slowly, it is limited to its site of occurrence with low invasive and metastatic capacity. Malignant tumors usually grow faster and invade the surrounding tissues with high capacity to metastasize. In the present review, canine mammary tumors were restricted to malignant phenotype, unless otherwise stated. This issue was clarified in Lines 51-52.

  1. Author list: ..., Zuccari and D.A.P.C. Remove "and"

Reply: Author list was corrected. Indeed, the word “and” was not inserted in the main text submitted for review, but edited along text processing by MDPI assistants.

  1. I would like to see a concise comparison of the clinicopathologic and incidence rate similarities between human and canine mammary malignancies (I assume you're not interested in the benign tumours ... or are you?).

Reply: We appreciate the reviewer suggestion and agree that such comparison would improve our manuscript. However, Editors and other reviewer both requested to remove any information regarding human breast cancer, emphasizing only canine mammary tumors. Therefore, we decided to follow their requirements for submission.

  1. Line 219 what is ObR?

Reply: ObR stands for leptin receptor. This information was inserted in the main text Line 218.

  1. Unless absolutely required, remove information about non-mammary cancers - it is confusing and makes the paper more difficult to read.

Reply: Information regarding non-mammary tumors was revised and removed as suggested.

  1. Spell out acronyms the first time you use them e.g. TNBC

Reply: Acronyms were properly addressed.

  1. Overall structure needs review - Section 2 seems to be a fairly concise summary of human mammary cancer metabolic information, Section 3 is about metabolic reprogramming in dogs and seems to be divided into several subsections but the naming is inconsistent. Section 3.1 is about carbohydrates Should Section 4 be renamed Section 3.2 Lipids, Section 5 should be Section 3.3 amino acids etc. Information in these sections is mixed up between humans and dogs : I would leave the dog information in place and delete the human information or summarise it back in Section 2 if it has not already been done.

Reply: Thank you for this suggestion. The text file uploaded during submission did not contain any Sections, just subtitles. The section numbering was performed by the MDPI assistant, and the revised text was corrected after your comment. Regarding information on human and dogs, we were requested by Editors to remove human data and leave only canine data in the manuscript. This generated the Section 2 which is an overview of the metabolic reprogramming in general, even though most of the citations refers to breast cancer because most of information in the literature was collected in humans. However, along the text you may eventually find information out of Section 2 about human breast cancer that was inserted to highlight either specific similarities or differences between species with an objective. However, we appreciate the reviewer’s suggestions and, unless necessary, human information was changed to Section 2 or removed.

  1. I noticed very little critical appraisal of the papers cited, at least some of which are based on poorly conducted or overinterpreted clinical investigations.

Reply: Thank you for your comment. To the best of our knowledge, this is the first review of its kind and was conceptualized after notice that there is a huge lack of information regarding the metabolic alteration in canine mammary tumors compared to human breast cancer. Also, that few available information was mostly descriptive without an elucidation of the underlying mechanism, or the sample size was very limited. Indeed, this could lead to misinterpretation, but our aim was to reunite the published information, open a debate about the issue and raise questions that were not fully addressed or remain elusive. This limitation was highlighted in the Conclusion Section. In addition, comments were added in the statements to highlight study limitations, as seen in Lines 360-380.

  1. Another point about referencing - refer to original work rather than other reviews e.g. Gray et al is not a good reference for the first sentence in the introduction - you should really find where that paper sourced information

Reply: References were checked, and reviews avoided unless absolutely convenient to the topic.

  1. at the outset in this manuscript, you should make it clear that when you state, for example, in line 1 "the most common cancer in female dogs" you mean intact female dogs that have not been surgically neutered.

Reply: Thank you for your suggestion. Indeed, we referred to intact female dogs. This alteration was done as requested in Line 36.

  1. Conclusion - there are a lot of potential targets - what do you think are the two or three most promising for future investigation?

Reply: Thanks for your comment. It is hard to define what would be the promising targets, but up to date glutaminase and fatty acid synthase inhibitors have received efforts from researchers in human breast cancer. Since there are several metabolic alterations shared between humans and dogs, these could be new opportunities in canine mammary tumors as well. This consideration was inserted in the main text.

Reviewer 2

It presents a considerable number of references, but the authors do not mention the criteria used for research and article selection. Some recent contributions on the subject were not included in this review, such as:

Razzuoli, E., De Ciucis, C. G., Chirullo, B., Varello, K., Zoccola, R., Guardone, L., ... & Modesto, P. (2022). Molecular Characterization of CF33 Canine Cell Line and Evaluation of Its Ability to Respond against Infective Stressors in Sight of Anticancer Approaches. Veterinary Sciences, 9(10), 543.

Maués, T., Oliveira, T. F. D., El-Jaick, K. B., Figueiredo, A. M. S., Ferreira, M. D. L. G., & Ferreira, A. M. R. (2021). PGAM1 and TP53 mRNA levels in canine mammary carcinomas–Short communication. Acta Veterinaria Hungarica, 69(1), 50-54.

Reply: We appreciate your suggestions and both publications were added to the text. They can be found in Line 180 and in Lines 170-173, respectively. 

The text also cites studies on canine tumours other than mammary tumours, without clearly referring so. An example is the sentence in lines 198 and 199, including a research article on hemangiosarcoma along with the other two, both including results on canine mammary tumours. The same lack of precision is present in the manuscript concerning species; often it is not clear whether the authors are referring to human cancer or canine cancer such as in line 270, when mentioning prostate (cancer?). Considering that this is presented as a review on the metabolic changes present in CMTs, it would be expected that whenever resorting to refer studies on other types of tumours and species other than the canine, this should always be clarified.

Reply: We appreciate the reviewer suggestion. Regarding the citation in Lines 198-199 we would like to clarify that as you properly noticed, the study evaluated several cancers, including mammary. This is the reason why the work was mentioned. It is worth to emphasize that investigation on metabolic reprogramming of canine mammary tumors is not a very explored field yet, so we looked preferably for dedicated studies on canine mammary tumors but eventually we retrieved information from publications with more than one cancer type to make our conclusions. Concerning the species referring, it was revised, and all information was clarified or was not mentioned in case of well-known cell lines or other human model. In Line 270, the reference to prostate was to emphasize the antitumoral effects of the omega-3 which was the main point in the statement. Regardless of this clarification, the text was revised as requested and the issue addressed.

 A more serious flaw is the lack of precision when using terms such as “stage” (see lines 187 and 188). In Oncology, stage is not a synonym for malignancy or benignity and it should not be used as such. Cancer staging is different, and it is not to be confused with grade, either. Likewise, the term “subtype” seems to be used indifferently to refer to different histological subtypes, to malignant vs. benign tumours (see line 198, since the cited references describe how 18F-FDG-PET/CT may be useful to distinguish malignant and benign tumours, to detect metastasis and for tumour staging but it is inaccurate to consider it distinguishes histological subtypes). In line 355 it is said “high heterogenicity” is observed in “all CMT subtypes”, although the cited work by Kowal et al. is only on malignant mammary tumours (and not on benign CMTs, which have several histological subtypes too).

Reply: Thank you for your comment. In the present review, the term stage refers to distinct phases of cancer progression, such as from a less to a more aggressive tumor, which is different from grade, indeed. Here, we would like to clarify that stage was not used to designate benign or malignant status, and this distinction was emphasized when pertinent. Regarding the term subtype, we refer to it as the histological status of the malignant tumor considering the expression of estrogen receptor (ER), progesterone receptor (PR) or human epidermal growth factor receptor 2 (HER2/ERRB2). Using this classification, tumors, at least in humans, are classified as luminal, HER2 positive or triple-negative which has been compared to canine mammary gland tumors. Also, the term subtype is usually adopted in breast cancer classification and clinical reports, but also in canine mammary malignant tumors (PMID: 31807173). We apologize for any misunderstanding and revised the entire manuscript to address this issue, including a clear designation of each term in Lines 51-55.

 Overall,how this work is written lacks clarity and English should be extensively edited.

Reply: The main text was revised for a native speaker and edited following all the reviewers’ recommendations.

Sometimes a phrase begins, and it is not clear what the subject is. An example of this (lines 41 to 45):

“Sexual hormones deregulation, such as exposure to endogenous ovarian hormones, may cause the mammary tumors development in dogs [3–5], but others may also influence CMT such as obesity in early stages of life [6], inflammation [7] and the increase in free radicals and reactive oxygen species (ROS) [8]. They may appear as single or multiple nodules, and posterior…”

 To whom does the plural subject “They” refer? To sexual hormones? To ROS? There are multiple similar phrases throughout the text. In l,nes 162-163 it is said “considerably [66]. This has been linked to higher lactate production and release by cancer cells, a well-known feature of the Warburg effect, which is converted to glucose through gluconeogenesis in the liver.”. As it is, the sentence suggests that the Warburg effect is converted to glucose through gluconeogenesis by the liver.

Reply: We sincerely appreciate your feedback. The manuscript has been carefully revised, and the statements have been corrected accordingly. We would like to clarify that in Lines 41-45, the subject "They" refers to CMT, as any of the other objects mentioned in the statement could have nodules as well. Regarding Lines 162-163, we intended to convey that lactate is converted to glucose through gluconeogenesis in the liver, which is a well-known process. However, to prevent any potential misunderstanding by non-experts in the field, we have rewritten the sentence as suggested.

There are also many long sentences, lacking punctuation and some phrases that seem unfinished (see, for example, lines 303 and 304) or that need to be re-written (see lines 324 and 325).

Reply: The main text was entirely revised and sentences shortened as requested.

The term “hormone positive cells” could probably be replaced by “hormone receptor positive cells”.

Reply: The term “hormone positive cells” was changed to “hormone receptor positive cells”, as requested.

TNBC is an abbreviature, but it is never explained in the text to what it refers to.

Reply: The acronym TNBC was explained in its first appearance in Line 108.

Some conclusions may be considered quite bold such as when it is said “Therefore, this evidence shows that, from the metabolic standpoint, the amino acid profile is altered in CMT and that glutamine metabolism seems to follow the same response in dogs and humans. “, as the authors on the following sentence write “ Although further studies are required to elucidate this issue, this is an important finding (…)”. If it is well described that the amino acid profile is altered in CMTs, saying there is evidence that glutamine metabolism follows the same response in dogs and humans when CMTs are yet to be fully characterized concerning this aspect may be a bit forceful.

Reply: Thank you for expressing your concern. However, we respectfully disagree with the reviewer's characterization of our statement as bold or forceful. In order to address any potential misinterpretation, we took care to emphasize in the text that further investigation is needed to confirm the association between glutamine metabolism and the observed response.

The artwork is adequate but the addition of a table summarizing the findings exclusively in canine mammary tumours, both in vivo and in vitro, with cited sources, would be a welcome addition.

Reply: We sincerely appreciate the suggestion made by the reviewer. However, we would like to kindly point out that Figure 2 already provides a comprehensive summary of the main findings, both in vitro and in vivo, specifically focusing on canine mammary tumors. As per the guidelines set by MDPI for review papers, it is required to include a minimum of two figures. Converting Figure 2 into a table would result in unnecessary redundancy.

Round 2

Reviewer 2 Report

I congratulate the authors for the much-improved manuscript now presented.

Although the authors chose not to detail and highlight all the changes in the revised text and in their reply, most of the situations were addressed, irrespective of the written reply. Such was the case of corrections such as "prostate cancer" instead of "prostate", or the choice to eliminate reference to the paper by Borgatti et al. (2017), which was focused only on hemangiosarcoma (and not CMTs) or the inclusion in line 108 of the significance of TNBC, absent in the previous version.

I also welcome the clarification of the author's use of terms such as grade, subtype, and benign and malignant. However, the use of the term stage is still not aligned with what is staging in oncology.

According to the World Health Organization, and in Oncology practice, when staging cancer, elements such as primary tumor site, tumor size, number of tumors (multiplicity), depth of invasion and extension to regional or distant tissues, involvement of regional lymph nodes, and distant metastasis. This is staging cancer, as it is understood by WHO, and by specialist associations such as the National Cancer Registrars Association. Staging is used to establish a prognosis and guide therapeutic choices in clinical practice.

The staging process varies with the type of tumor considered; for example, in canine mast cell tumors, cytology of peripheral blood buffy coat should be included, while this procedure is not deemed necessary in canine mammary tumors.

Staging systems in canine mammary tumors include the widely and historically used TNM, as nicely described by Sorenmo et al. (2012), and new developments such as described by Chocteau et al. (2019), and the authors may find these works useful.

Sorenmo KU, Woerley DR, Goldschmidt MH. Tumors of the mammary gland. In: Withrow SJ, Vail DM, Page RP, editors. Withrow and MacEwen’s Small Animal Clinical Oncology, 5th ed. Philadelphia, PA: Saunders Company (2012). p. 538–556. doi: 10.1016/B978-1-4377-2362-5.00027-X

Chocteau, F., Abadie, J., Loussouarn, D., & Nguyen, F. (2019). Proposal for a histological staging system of mammary carcinomas in dogs and cats. Part 1: canine mammary carcinomas. Frontiers in veterinary science6, 388.

And this article, on staging human breast cancer

Teichgraeber, D. C., Guirguis, M. S., & Whitman, G. J. (2021). Breast cancer staging: Updates in the AJCC cancer staging manual, and current challenges for radiologists, from the AJR special series on cancer staging. American Journal of Roentgenology217(2), 278-290.

"The authors say in line  52 that “Stage” the term used to designate an initial or advanced step in cancer progression, usually related to aggressiveness"

In fact, a malignant tumor may be of very high grade (very aggressive), although small in dimension and not metastatic. So, a tumor may be very aggressive locally, locally invasive, and with short duplication times (due to being high grade) and still be classified in a lower stage than another tumor, that has lower grade but shows metastasis or regional lymph node infiltration, for example. 

When the authors write in line 234 to 237 "Carnitine Acylcarnitine Translocase (CACT), Car-234 nitine Palmitoyl transferase 2 (CPT2), and Carnitine O-acetyltransferase (CrAT) were found to be overexpressed in CMTs in cell lines and tissue compared to normal condition, except for a decrease observed in a poorly undifferentiated stage" really refers to what Cacciola et al. describe correctly as CACT decreased expression in poorly differentiated carcinomas G3 (higher grade). Stage and grade are different things. A poorly differentiated tumor population is further distanced from the original healthy population and often shows very aggressive biological behavior.

When addressing metabolic changes associated with tumors and establishing relations with tumor aggressiveness it is important to correctly interpret the cited sources when they say very aggressive, higher-grade tumors show decreased expression. 

Cacciola et al. (2021) wrote:

"CACT expression was negatively correlated with tumor malignancy, with CACT expression downregulated in poorly differentiated G3 carcinomas compared to NMGs, well-differentiated (G1) and moderately differentiated (G2) carcinomas.(...)  Consistent with our findings that CACT expression was downregulated in CMTs, another previous study confirmed that in human bladder cancer patients, the expression of CACT was significantly deregulated in cancer tissues compared with healthy bladder tissues"

I would suggest the replacement of the word "stage" in line 237 with "poorly differentiated, higher grade CMTs".

The use of the term "stage" in line 270 "stage III lymphoma" is correct and informs the reader that the study encompassed lymphomas with generalized lymph node involvement and without systemic signs. It does not inform on the type of lymphoma or its aggressiveness (is the cited study, these were high grade, lymphoblastic lymphoma).

Figure 2 summarizes the metabolic alterations in canine mammary tumors. However, CMTs are highly heterogeneous histologically and in terms of biological behavior. Thus, it would not be redundant to clarify whether the resumed alterations were described in vitro in established cell lines, in primary cultures, in vivo, etc, since this information is thus not present in the figure, which is helpful and informative. It was never suggested Figure 2 to be replaced. However, the systematic presentation of findings and respective sources in an additional table is still suggested.
